# Examining the Potential of Vitamin C Supplementation in Tissue-Engineered Equine Superficial Digital Flexor Tendon Constructs

**DOI:** 10.3390/ijms242317098

**Published:** 2023-12-04

**Authors:** Michael J. Mienaltowski, Mitchell Callahan, Nicole L. Gonzales, Angelique Wong

**Affiliations:** 1Department of Animal Science, College of Agricultural & Environmental Sciences, University of California Davis, Davis, CA 95616, USA; 2School of Veterinary Medicine, University of California Davis, Davis, CA 95616, USA

**Keywords:** equine, tendon, peritenon, vitamin C, three-dimensional construct

## Abstract

Because equine tendinopathies are slow to heal and often recur, therapeutic strategies are being considered that aid tendon repair. Given the success of utilizing vitamin C to promote tenogenesis in other species, we hypothesized that vitamin C supplementation would produce dose-dependent improvements in the tenogenic properties of tendon proper (TP) and peritenon (PERI) cells of the equine superficial digital flexor tendon (SDFT). Equine TP- and PERI-progenitor-cell-seeded fibrin three-dimensional constructs were supplemented with four concentrations of vitamin C. The gene expression profiles of the constructs were assessed with 3′-Tag-Seq and real-time quantitative polymerase chain reaction (RT-qPCR); collagen content and fibril ultrastructure were also analyzed. Moreover, cells were challenged with dexamethasone to determine the levels of cytoprotection afforded by vitamin C. Expression profiling demonstrated that vitamin C had an anti-inflammatory effect on TP and PERI cell constructs. Moreover, vitamin C supplementation mitigated the degenerative pathways seen in tendinopathy and increased collagen content in tendon constructs. When challenged with dexamethasone in two-dimensional culture, vitamin C had a cytoprotective effect for TP cells but not necessarily for PERI cells. Future studies will explore the effects of vitamin C on these cells during inflammation and within the tendon niche in vivo.

## 1. Introduction

Tendinopathies are perhaps the most common musculoskeletal injury in horses of all uses [1]. Slow recovery and recurring injury of tendons often lead to the retirement of horses from competitive performance [1,2]. Furthermore, the superficial digital flexor tendon (SDFT) is the tendon that is most commonly injured, with 75–93% of tendon injuries being diagnosed in the SDFT [1,2,3]. The SDFT is an energy-storing tendon that is highly susceptible to the accumulation of microdamage from extreme loading environments experienced from the high stresses induced by exercise [4]. As microdamage accumulates, degeneration occurs in the tendon such that there are increases in cellularity, in the levels of sulfated glycosaminoglycans, and in levels of type III collagen content, with decreases in collagen crosslinking—all characteristics associated with a repair response [5]. Thus, tendinopathies result from acute and/or chronic stresses that ultimately weaken tendon structure, consequently affecting tendon function. Therefore, stimulating proper tendon formation, upholding or fortifying mature tendons, and fostering effective repair are essential for promoting the musculoskeletal well-being of horses.

The structure and function of a tendon like the SDFT depend upon tendon cells and the connective tissue matrix that they assemble [6,7,8,9,10]. The cells of the tendon proper (TP) and the cells of the peritendinous (PERI) tissue are important for tendon development, maturation, homeostasis, and repair [7,8,11]. Within the peritenon and tendon proper exist several cell populations, including terminally differentiated tenocytes, tenoblasts, which are believed to have a more progenitor-like status, endothelial based cells, immune cells, and pericytes [12]. Peritenon cells are predominant in repair, though they are less capable of forming tendon connective tissue, while tendon proper cells are less active in repair though they have greater tenogenic capabilities [8,13,14,15]. Recently, the roles of progenitor cells of the tendon proper and peritenon have been considered because these cells could be targeted as a source for autologous stem/progenitor cells to be used in repair, and therapeutic strategies could be focused on stimulating increased activity in situ to promote repair [16,17,18]. Of course, equally important to the cells of the tendon are the proteins that they secrete that are responsible for matrix organization—fibrillar collagens and accompanying regulatory molecules [9,10]. Thus, when considering tendon repair strategies, it is important to consider how both TP and PERI cell populations in the tendon niche might respond to strategies that bolster tenogenesis.

Vitamin C, or ascorbic acid, is an essential cofactor required for the hydroxylation of prolines to hydroxyprolines and lysines to hydroxylysines [10]. Hydroxyprolines and hydroxylysines contribute to the stability of the triple helices of each collagen molecule [19]. Because vitamin C plays an important role in collagen fibrillogenesis, several researchers have examined its utility in tendon repair. For example, when vitamin C was injected into experimental flexor tendon injuries in chickens, fibrosis and adhesions were reduced; in one study, vitamin C treatment was associated with increased levels of reduced glutathione (GSH) available as an antioxidant [20]. Moreover, when rats with experimentally induced Achilles tendon injuries were treated with intraperitoneal injections of vitamin C, collagen synthesis increased and overall collagen fiber diameters increased with healing as the tendons showed an improved maximal tensile load [21,22]. Furthermore, when vitamin C was applied to human tenocytes in vitro, cell proliferation, cell viability, cell migration, and collagen accumulation improved [23,24,25]. Moreover, rabbit Achilles tenocytes supplemented with vitamin C had increased expression of the tenogenic transcription factor Mohawk, as well as increased levels of type I collagen genes, the extracellular matrix (ECM) assembly regulator SLRP DCN, and Matrix metalloproteinase-2 which is active in healing [26]. Based on these observations, we hypothesized that supplementation of vitamin C would produce dose-dependent improvements in the tenogenic properties of peritenon and tendon proper cells of the equine SDFT. Such improvements include the expression of markers of tenogenic differentiation and increased deposition of collagen. To test this hypothesis, we examined the response of TP and PERI progenitor cells to several doses of vitamin C in an established three-dimensional culture [27]. We isolated equine SDFT progenitor cells based on a protocol established originally by Bi et al. [28], which we then used for a mouse Achilles tendon [7,8,11]. We used transcriptomics to investigate which genes and pathways were activated by vitamin C in each cell type. In addition, we examined how vitamin C affected collagen content within tendon constructs, as well as the effects of its supplementation on tendon construct ultrastructure. Finally, we examined the cytoprotective effects of vitamin C on TP and PERI progenitor cells challenged with dexamethasone as corticosteroid treatments may sometimes be used in cases of tenosynovitis and navicular disease [29,30]. Doses of vitamin C were selected in increments of 200 μM from 0 to 800 μM. A dose of 200 μM ascorbic acid is commonly used in the tendon construct model [27,31], while 0 μM has never previously been tested on equine cells in this model. The plasma level of ascorbic acid in adult horses is about 33.5 μM [32]; the age-adjusted mean concentration of plasma ascorbic acid in humans is 51.4 μM [33].

## 2. Results

### 2.1. Transcriptomic Assessment of Vitamin C Supplementation

Next-generation sequencing was performed on each sample, considering horse (*n* = 4) × cell type (*n* = 2, TP and PERI) × vitamin C concentration (*n* = 4; 0 μM, 200 μM, 400 μM, and 800 μM). Profiling with 3′-Tag-seq provided a robust ~5 million reads per sample with 91.7–95.1% of reads mapped to the EquCab3 reference genome (Table 1). Considering a *p* ≤ 0.05, 182–328 genes were differentially expressed. After applying a false discovery rate (FDR) of *q* ≤ 0.20, only a handful of differentially expressed genes were included for consideration (Table 2). To further understand transcriptomic profiles across the samples and treatments, we followed a screening strategy to assess expression trends from the supplementation of vitamin C, as previously described [11,34]. Using the *p* ≤ 0.05 threshold, there were 35 genes for TP progenitors and 65 genes for PERI progenitors, for which there were shared expression profiles for the vitamin C-treated (200 μM, 400 μM, and 800 μM) group vs. the untreated control (0 μM) (Figure 1, Appendix A). When the shared genes were examined within the NIH DAVID Functional Annotation Tool, overrepresented functional annotation was noted; for TP cells in constructs treated with vitamin C, the shared upregulated genes represented “collagen-binding” and “secreted” annotations (Appendix A). For PERI cells in constructs treated with vitamin C, the shared downregulated genes were overrepresented for functional annotations of “myosin”, “myosin complex”, “muscle protein”, and “calmodulin-binding” (Appendix A).

When examining DEGs amongst the groups of constructs depending upon cell type and vitamin C concentration, relative to their respective control constructs and within the *q* ≤ 0.20 threshold, differential expression was noted for several genes (Table 3). When PERI-cell-derived constructs were supplemented with 200 μM vitamin C, upregulated DEGs included those that are expressed at the tendon insertion (*GPC3*) [35] and encode an anti-inflammatory protein (*TNFAIP3*) [36]. The supplementation of 200 μM vitamin C to PERI-cell-derived constructs also led to the relative downregulation of genes that encode pro-inflammatory proteins (*CCL13*, *IRF8*), a proteoglycanase (*ADAMTS1*), a growth factor that contributes to chondrogenesis and osteogenesis (*BMP4*), a fibrotic tissue marker (*IL11*), and a myosin protein (*MYH7*). Likewise, when PERI-cell-derived constructs were supplemented with 800 μM vitamin C, *MYH7* was downregulated, as was the matrixmetalloproteinase-encoding gene *MMP12*, which is a macrophage metalloelastase commonly implicated in atherosclerosis [37,38]. Trends for increased expression of *TNFAIP3* and decreased *BMP4* expression with vitamin C supplementation of PERI constructs were seen in RT-qPCR as well (Figure 2A,C), while the expression of *MMP12* was only confirmed to trend downward with supplementation of 800 μM vitamin C (Figure 2B).

DEGs were also noted for TP-cell-derived constructs supplemented with vitamin C. When TP constructs received 200 μM vitamin C, upregulated DEGs included *DDX60L* and *IFI44*, which have been found to be upregulated in tenocytes and fibroblasts in the human semitendinosus muscle–tendon complex, though their roles are still unclear [39]; glycosaminoglycan (GAG)-binding *CXCL8* was downregulated [40]. The gene *SMOC2*, which has been shown to be expressed in tendon fibroblasts, was upregulated in TP constructs receiving 400 μM vitamin C [12,41]. In contrast, *CXCL6* and *CXCL8* pro-inflammatory genes and an active marker of inflammation, *SAA1*, were downregulated with 400 μM vitamin C supplementation. The gene *JAM2*, which has a role in mesenchymal stem cell motility, was upregulated in TP constructs receiving 800 μM vitamin C [42]. Trends for decreased expression of *CXCL8* and increased *DDX60L*, *IFI44*, and *SMOC2* expression with vitamin C supplementation of TP constructs were seen in RT-qPCR with significant increases for *DDX60L* (800 μM) and *SMOC2* (400 μM) genes (Figure 2D–G).

RT-qPCR was also used to examine the expression of genes commonly used to describe the tenogenic status of cells; while no significant differences were reported, trends in expression were found. The expression of *SCX* and *COL1A1* trended higher for TP constructs receiving vitamin C relative to control TP constructs (Appendix A). The expression of *MKX* trended higher for TP constructs receiving 200 μM vitamin C (Appendix A), and the expression of *BGN* and *DCN* trended higher for TP constructs supplemented with 200 μM and 800 μM vitamin C (Appendix A). For PERI constructs, *SCX*, *BGN*, and *DCN* expression trended higher for those constructs receiving 200 μM and 400 μM vitamin C (Appendix A).

### 2.2. Comparisons of Transcriptomic Profiles of TP versus PERI-Cell-Derived Constructs

Several studies have demonstrated that TP and PERI cells respond differently to tendon repair in vivo and to other stimuli in vitro [8,14,15]. DEGs were also examined in comparisons of TP- and PERI-cell-derived constructs for the four concentrations of vitamin C provided. There were 26–60 DEGs between TP and PERI constructs, depending upon vitamin C concentration based on a *q* ≤ 0.05 threshold (Table 4). DEGs with increased abundance in TP or PERI were applied to the NIH DAVID Functional Annotation Tool, and overrepresented gene ontology categories were determined for sets of genes by cell type and vitamin C concentration (Table 4). “Extracellular space” or “extracellular region” were common cellular compartment ontologies found across DEGs. There was also a pattern of increased expression of “heparin binding”, “chemokine activity”, and “CXCR chemokine receptor binding” for PERI-cell-derived constructs. When comparing TP vs. PERI expression profiles for constructs supplemented with vitamin C, 19 DEGs were shared amongst all three vitamin C concentrations, 200 μM, 400 μM, and 800 μM (Figure 3). Of the 19 DEGs, 11 genes were more abundant in TP-derived constructs, and 8 genes were more abundant in PERI-derived constructs (Figure 4). Again using the NIH DAVID Functional Annotation Tool, the TP-cell-derived constructs had genes in the “extracellular region” (*AEBP1*, *IGFBP7*, *MGP*, *MMP13*, *SLIT2*) and “developmental protein” (*MGP*, *SFRP2*, *SLIT2*) ontologies, while the PERI-cell-derived constructs had genes in the “chemokine activity” (*CXCL1*, *CXCL6*, *CXCL8*) and ”extracellular space” (*CXCL1*, *CXCL6*, *CXCL8, SFRP1, SERPINB10*) ontologies.

### 2.3. Collagen Content

A hydroxyproline assay was used to determine collagen content within the TP- and PERI-progenitor cell-derived constructs supplemented with vitamin C. As these constructs were originally composed of cells and fibrin, any collagen in the construct was secreted by the cells. For TP cells, there was a trend of increasing collagen content with vitamin C supplementation (Figure 5A). For PERI cells, there was a significant increase in collagen content for PERI cells receiving 400 μM vitamin C with trending increases for concentrations of 200 μM and 800 μM (Figure 5B).

### 2.4. Collagen Fibril Diameters from Transmission Electron Microscopy

Ultrastructural analyses of the TP- and PERI-cell-derived constructs were performed using transmission electron microscopy. TP constructs receiving 400 μM and 800 μM vitamin C had smaller mean fibril diameters relative to those constructs receiving 0 μM and 200 μM vitamin C (Figure 6A). In looking at the distribution of fibril diameters in the TP constructs receiving 400 μM and 800 μM vitamin C, there were bimodal distributions containing fibrils of a diameter similar to that of the majority of fibrils across all TP constructs (40–50 nm) and a subset of smaller fibrils (~30 nm) (Figure 6C). In contrast, for PERI-cell-derived constructs, mean fibril diameter increased in a vitamin C-dose-dependent manner (Figure 6B). Bimodal distributions were also noted for PERI constructs at all levels of vitamin C supplementation with the peaks at each level representing increasingly larger fibril diameters (Figure 6D).

### 2.5. Dexamethasone Challenge

When TP and PERI progenitors were challenged in two-dimensional culture with dexamethasone for 3 days, the number of surviving cells decreased for both cell types at doses of 5 nM and 10 nM (Appendix A). The 10 nM challenge was repeated but with co-supplementation with 0 μM, 200 μM, 400 μM, or 800 μM vitamin C, and apoptotic activity was measured instead. We saw that apoptotic activity was decreased for TP cells, particularly at 400 μM supplementation (Figure 7A) while PERI cells were not rescued from apoptosis by vitamin C supplementation at any level (Figure 7B).

## 3. Discussion

The supplementation of vitamin C had beneficial effects on SDFT TP and PERI progenitor cells. Transcriptional profiling of the SDFT TP- and PERI-cell-derived constructs demonstrated that vitamin C exhibited anti-inflammatory, calcification inhibitory, cell-proliferation-promoting, and/or pro-tenogenic activities at various concentrations for each cell population. Collectively, for TP cells, vitamin C promoted the transcriptional expression of genes encoding extracellular matrix molecules involved with collagen, while it appears that for PERI cells—many originating from vasculature—vitamin C contributed to the reduction in expression of genes encoding muscle proteins. Thus, one could interpret these findings as indicating that vitamin C promoted a tenogenic expressional profile in TP cells and lessened the non-tenogenic expressional profile found in PERI cells. Previous studies have demonstrated that cells from each region have differing tenogenic capacities in vitro in three-dimensional culture and in vivo during tendon repair [8,14,15]. Even with the addition of vitamin C with the in vitro constructs, the regional differences between TP and PERI cells could not be reconciled.

Based on the transcriptomics results, vitamin C demonstrated anti-inflammatory properties. At 200 μM, PERI cells had increased levels of *TNFAIP3* and decreased levels of *CCL13*, *IL11*, and *IRF8*. At 200 μM, TP cells had reductions in the levels of *SAA1* and *CXCL8*; at 400 μM, the levels of *CXCL6* and *CXCL8* were lowered in TP cells. TNF-alpha-induced protein 3 (*TNFAIP3*), which had the greatest level of expression in PERI cells treated with vitamin C, acts to inhibit TNF-induced, TLR-induced, and IL-17-induced NF-kB pathways [43]. TNFAIP3 can also inhibit the transcription of *NLRP3*, *ASC*, *procaspase 1*, *pro-IL-1β*, and *proIL-18*, which upon activation of NF-kB contribute to inflammation, the secretion of IL-1β, and the activation of pyroptosis [43]. Additionally, vitamin C supplementation led to decreases in the expression of the chemokines *CCL13*, *CXCL6*, and *CXCL8*. A reduction in such chemokines is beneficial for modulating chronic inflammation [44,45]. Likewise, decreasing levels of *IL11* could be beneficial for mitigating fibrosis caused by vascular-type PERI cells, which likely undergo vascular smooth muscle cell phenotypic switching in a tendon repair setting because they are not truly tenocytes [46,47]. Similarly, the gene *Serum amyloid A1* (*SAA1*)—which encodes a major acute-phase protein responsible for activating fibroblasts involved in pathological repair and scarring as well as working synergistically with chemokines to recruit monocytes to the injury—was decreased with vitamin C supplementation [48,49]. Thus, the expression data indicated that vitamin C modulated genes that act within multiple inflammatory pathways and processes for equine TP and PERI cells.

When a horse is being treated for tendinopathy, interventions may also focus on the prevention of degenerative progression of pathology. In this study, vitamin C was able to mitigate matrix degradation and tendon tissue calcification. For PERI cells, vitamin C reduced expression of *ADAMTS1*, which is a protease that has been shown to be active in degradative processes of tendinopathy [50,51]. Reduced expression of *MMP12* was also demonstrated; this macrophage metalloelastase is present in both inflammatory conditions requiring macrophage migration and instances of extracellular matrix remodeling, as in repair and in response to loading [38,52,53]. The local inhibition of specific proteases that might degrade networks of fibrillar collagens which are essential to tendon function has been suggested as a potential intervention for tendinopathies [54]. Moreover, the expression patterns of *BMP4* and *IFI44* for treated PERI cells suggest that vitamin C supplementation inhibits chondrification and ossification in these cells [55,56,57]. Thus, the results from expression profiling demonstrate the potential of vitamin C to slow degenerative progress.

Besides halting the progression of disease, ultimately it is important that any intervention for tendinopathy promotes a tenogenic phenotype. Such a phenotype includes the expression of transcription factors like Scleraxis and Mohawk and matrix assembly molecules like Decorin, as well as the production and organization of fibrillar Collagen I. Previously, human and rabbit tenocytes treated with vitamin C demonstrated increased expression of genes like *SCX*, *MKX*, *DCN*, and *COL1A1* [26,58]. In this study, supplementation with vitamin C increased the expression of all such markers for the tendon phenotype, though not with statistical significance. Furthermore, when comparing TP and PERI constructs, it appears that the TP cells tended to have greater levels of gene expression for these markers, though again, statistical significance was not achieved. That said, we did note increases in collagen content for the TP- and PERI-cell-derived constructs treated with vitamin C, which has also been the case in rat injury models [22,59] and in human ligament models given plasma from athletes supplemented with vitamin C [60]. In examining the ultrastructure of the tendon constructs using TEM, we saw a dichotomy in the effects of vitamin C supplementation. TP-cell-derived constructs had a slightly smaller mean fibril diameter due to increased numbers of smaller-diameter fibers, thus suggesting the deposition of new fibrils over the period of culture. In contrast, the fibril diameters of PERI-cell-derived increased in diameter, which suggests that the fibrils generated were maturing to thicker fibril diameter sizes over the period of culture. Still, within the in vitro construct model, collagen fibril diameters did not approach the ~70 nm diameter mean seen in vivo for adult equine SDFTs [61].

The role of vitamin C as an antioxidant has been well defined [62,63]. Its role as a potent antioxidant in tendons was confirmed when treating human tenocytes [23]. Moreover, vitamin C has been found to rescue deficits in superoxide-dismutase-1 in the degeneration of aging *Sod1*^−/−^ mouse rotator cuffs [64]. Similarly, when rats received vitamin C post-operatively when undergoing infraspinatus insertion repair, oxidative stress biomarkers were significantly decreased, and SOD activity was upregulated [65]. Interestingly enough, in this study, differential expression of antioxidant genes was not predominant for the TP and PERI cells. However, transcriptome analysis was not performed under noxious or injurious conditions (e.g., inflammation or toxicity). That said, when we challenged TP and PERI cells with 10 nM dexamethasone in two-dimensional culture, vitamin C did reduce levels of apoptotic activity in the TP cells, particularly at a concentration of 400 μM. Thus, much like with human tenocytes, vitamin C had cytoprotective properties [66]. Vitamin C did not seem to have the same protective effect on PERI cells treated with dexamethasone; however, this could be related to the perivascular origins of PERI cells. That is, PERI cells have a pericyte phenotype [8,11] and could thus rely on closer communication with other cells in vasculature [67]. Because cells of the peritenon are responsible for much of the healing process in tendon injury, further studies are required to better understand how vitamin C affects cells within each region of the tendon in vivo, within the complete niche of the SDFT, even in the face of inflammatory and toxicity challenges.

In our previous studies, we found that TP and PERI progenitor cells have unique tenogenic properties in mice [8,11], and TP and PERI cells grown in tenocyte media respond uniquely to potential small leucine-rich repeat proteoglycans and adipose-derived stem cell therapeutic strategies in horses [35,68]. In this study, we also noted that progenitor cells from each tendon region had differences in their responses to supplementation of vitamin C. As concentrations of vitamin C were increased, we noticed that there was still greater expression of genes from immunity- and inflammation-related pathways for PERI cells relative to TP cells. Thus, this response suggests another unique feature of the TP cells; that is, expression of inflammation and immunity genes is generally lower in TP cells, and expression of these genes in PERI cells could still require further modulation to reach levels as low as those in TP cells. Again, while this study demonstrated that the equine SDFT TP and PERI progenitors responded to vitamin C supplementation, a limitation of this study is that we did not examine the transcriptional profiles of TP and PERI progenitors after a pro-inflammatory stimulus. Thus, follow-up studies are necessary to understand how vitamin C might affect these cells in inflammation.

This study had several other limitations. While collagen content was measured in the constructs, there could be great value to examining the activity of matrix metalloproteinases (MMPs) and tissue inhibitors of metalloproteinase to understand how vitamin C might affect the enzyme activity associated with collagen turnover, especially in follow-up studies in which these constructs are exposed to inflammatory conditions. Moreover, while our RT-qPCR phenotyping efforts discerned expression levels for stem cell markers defined by the International Society for Cellular Therapy (ISCT) and demonstrated the same adherence to plastic as other tendon progenitor studies [7,8,11,28,69,70,71], we did not discern each isolated progenitor cell set in trilineage assays because cells not classified as progenitors in the SDFT were previously demonstrated to be multipotent. Thus, our classification of these cells as progenitor cells is based on the RT-qPCR phenotype and previously described progenitor cell isolation technique [7,8], and further scrutiny of the cells in the SDFT is required. For example, there would be great benefit to performing single-cell profiling of all cells in the SDFT with further comparative analyses of these cell populations to those clonogenic cells in the SDFT.

## 4. Materials and Methods

### 4.1. Progenitor Cell Isolation

Superficial digital flexor tendon (SDFT) samples were harvested from four horses (Thoroughbreds and Quarterhorses ages 8–14 years; two mares and two stallions) that were euthanized for reasons outside of this study and were thus exempt from approval of the University of California Davis Institutional Animal Care and Use Committee. The horses had no known history of tendinopathy and were visually confirmed to be sound prior to euthanasia. Immediately upon euthanasia, sterile instruments were used to harvest 2.5 cm of the superficial digital flexor tendon located 10 to 15 cm proximal to the forelimb fetlock. Upon removal of the tendon, it was rinsed three times in Dulbecco’s Phosphate Buffered Solution (DPBS, Life Technologies, Benicia, CA, USA) containing 1% antibiotic/antimycotic before transport for further isolation [7,27]. Tendon samples were isolated into peritenon (PERI) and tendon proper (TP) regions sterilely under a dissecting microscope where 2–2.5 mm of the central tendon core was used for the tendon proper and the peritenon consisted of the viscous paratenon in addition to 1 mm of the epitenon. Separated tendon proper and peritenon regions were enzymatically digested following previous protocols using 0.3% type-I collagenase (CLS-1, Worthington, Lakewood, NJ, USA) and 0.4% Dispase II (Roche, Basel, Switzerland) in Hanks Balanced Salt Solution (HBSS, Gibco, Benicia, CA, USA) and inactivated following agitation in tendon progenitor media (α-MEM, 20% fetal bovine serum (FBS), 100 mM 2-mercaptoethanol, 2 mM L-Glutamine, 100 U/mL penicillin, 100 μg/mL streptomycin, 250 ng/mL amphotericin B) [8,28,31]. From both digests (peritenon and tendon), cells were strained with a 70 μm cell strainer. Cells were collected by centrifugation at 500× *g* for five minutes, resuspended in media, and counted using a hemocytometer with Trypan blue staining. Tendon and peritenon cells were plated in tissue culture flasks at 40 cells and 320 cells per cm^2^ in progenitor media so that adhering progenitor cells formed segregated colonies within the flasks as previously described [7,8,11,28]. Cells were expanded for as many as 14 days in T75 tissue-culture-treated flasks and passaged to P2 at 90% confluence thereafter before cryopreservation in 10% dimethyl sulfoxide (DMSO) in progenitor media and maintained in liquid nitrogen until used for this study. RT-qPCR profiling of both types of P2 progenitor cells was performed. A phenotype of *CD29*+/*CD34*−/*CD44*+/*CD45*-/*CD79A*−/*CD105*+ was seen for cells from both regions; TP progenitors demonstrated greater transcript levels of *CD90* and *CD105* (Appendix A). Transcript levels of *SCX* trended higher for TP cells. All horse samples were maintained separately to ensure independent biological replicates.

### 4.2. Constructs with Vitamin C Treatment

TP and PERI progenitor cells were thawed and plated at 1 × 10^6^ cells per T75 tissue-culture-treated flask in α-MEM, 10% fetal bovine serum, 2 mM L-Glutamine, 100 U/mL penicillin, 100 μg/mL streptomycin, and 250 ng/mL amphotericin B (basal media). Once cells reached 80% confluence, they were dissociated from flasks with 0.25% trypsin for 10 min and seeded at 3.0 × 10^5^ cells/construct within a fibrinogen and thrombin provisional matrix gel (1 mL containing cells, basal media, 5.8 U bovine plasma thrombin, and 5.72 mg of bovine plasma fibrinogen) atop a silicone elastomer between standardized brushite pegs set 10 mm apart [8,27]. TP and PERI constructs were cultured for 12 days. All TP and PERI constructs received media changes three times weekly with α-MEM, 10% FBS, 2 mM L-glutamine, antibiotics, and antimycotics [8,27]. Vitamin C (as ascorbic-2-phosphate) was supplemented in the media with constructs receiving 0 µM, 200 µM, 400 µM, or 800 µM.

### 4.3. 3′-Tag-seq

Tendon constructs for each horse and treatment group were homogenized using a BioSpec Tissue-Tearor in TRIzol. The aqueous phase of the TRIzol separation was then applied to the RNeasy Plus Micro Kit (QIAGEN, Valencia, CA, USA) including a RNase-free DNase treatment (QIAGEN, Valencia, CA, USA), as has been previously done [31]. RNA sample integrity was assessed using a LabChip GX (software version 5.3.2115.0) to assign RNA quality scores with the UC Davis Genome Center; quality scores for all samples were between 9.1 and 10.0. 3′ Tag RNA-seq libraries were generated from each sample with the UC Davis Genome Center from RNA (>500 ng in 20 µL) and sequenced on an Illumina HiSeq 4000 (Illumina, San Diego, CA, USA), generating 10 M single-end 90 bp reads per sample [72,73]. Data obtained with 3′-Tag-seq were processed using CLC Genomics Workbench 21 (QIAGEN). Quality control assessment was completed with QC for Sequencing Reads 0.2 based on FastQC. Then, reads were trimmed with Trim Reads 2.5 within Workbench (parameters: quality limit = 0.05; trim ambiguous nucleotides = yes; maximum number of ambiguities = 2) with the first 12 bp of the non-poly A reads trimmed to remove the random primer. PCR duplicates were removed. Single reads with a minimum of 20 bp were kept and aligned on the *Equus caballus* (Ensembl EquCab3.0) annotated reference genome using RNA-Seq Analysis 2.3 (parameters: mismatch cost = 2; insertion cost = 3; deletion cost = 3; length fraction = 0.8; similarity fraction = 0.8; strand-specific to both strands; maximum number of hits for a read = 10). Reads were normalized for sequencing and depth and reported as number of transcripts per million (TPM) reads. Differential gene expression was determined using differential expression for RNAseq 2.5 with a Wald test calculated from a general linearized model including false discovery rate multiple test correction, considering either treatment (200 μM, 400 μM, or 800 μM vitamin C vs. 0 μM; TP or PERI) or cell type (TP vs. PERI at each concentration). Differentially expressed genes (DEGs) were defined as *p* ≤ 0.05 or FDR *q* ≤ 0.20 for comparing concentrations and *q* ≤ 0.05 for comparing cell types. DEGs were applied to bioinformatics analyses. Venn diagrams were generated using CLC Genomics Workbench. Functional annotation of the DEGs was examined within The Database for Annotation, Visualization and Integrated Discovery (DAVID) [74,75].

### 4.4. RT-qPCR

For each sample representing horse x cell type x concentration, reverse transcription was performed on 600 ng total RNA using a High Capacity cDNA Reverse Transcription Kit (Life Technologies). Genes assessed by real-time quantitative polymerase chain reaction included *BMP4*, *CXCL8*, *DDX60L*, *IFI44*, *MMP12*, *SMOC2*, and *TNFAIP3*, the tenogenic differentiation genes *SCX* and *MKX*, and the ECM assembly genes *BGN*, *DCN*, and *COL1A1*. *POLR2A* was used as the housekeeping gene [11,35]. Genes used for stem cell marker profiling included *CD29*, *CD34*, *CD44*, *CD45*, *CD79A*, *CD90*, *CD105*, and *CD166* [70,71]. TaqMan primers were designed from equine gene structure annotation (NCBI Equicab 3.0) by submitting exons for genes for custom assays or from predesigned primers (Life Technologies). For RT-qPCR analysis, 1 µL of cDNA template was combined with TaqMan Fast Advanced Master Mix (Life Technologies) for a reaction volume of 20 µL in a StepOnePlus Real-Time PCR System (Applied Biosystems, Foster City, CA, USA) [31]. Each sample of amplified cDNA was analyzed in duplicate for each gene with gene specific efficiencies calculated using LinRegPCR v 7.5 software [76,77,78]. The relative quantity ratio formula was used to calculate the relative quantity of mRNA for each gene [76,78]. The RT-qPCR statistical analyses performed to validate expression of 3′-tag-seq findings were one-tailed Friedman’s non-parametric tests with Dunn’s multiple test correction. RT-The qPCR statistical analyses performed to examine expression of tendon markers were two-tailed non-parametric tests with Dunn’s multiple test correction.

### 4.5. Hydroxyproline Assay

Tendon constructs for each horse and treatment group were dehydrated on parchment paper for 20 min at 120 °C. Dried constructs were weighed and then added to 6 N hydrochloric acid at 120 °C for 2 h for hydrolysis, followed by 1.5 h to evaporate the hydrochloric acid [31,79]. Resulting pellets were then resuspended in hydroxyproline buffer (3.3% citric acid, 2.3% sodium hydroxide, 0.8% acetic acid in water, pH 6.0–6.5); stock samples were diluted 2:1 or 4:1 hydroxyproline buffer/stock sample to allow for more accurate colorimetric detection [31,79]. Finally, chloramine-T (14.1 mg/mL) and aldehyde perchlorate solution were added to each diluted sample before heating, cooling, and then measuring absorbance with a UV spectrophotometer at 550 nm, relative to hydroxyproline standards [31,79]. Analyses of hydroxyproline assays were carried out with one-way ANOVA with Dunnett’s multiple comparison test.

### 4.6. Transmission Electron Microscopy

Tendon constructs for each horse and treatment group were rinsed with phosphate-buffered saline (PBS) and fixed at length by complete immersion in Karnovsky’s fixative for 2 h at 4 °C before storage in transport solution for up to 1 week ahead of embedding [31]. Embedding was performed as has been previously described [7,80,81]. Briefly, fresh epoxy resin was used to embed constructs cut in half cross-sectionally and polymerized at 60 °C for 12 h (EMBed—812, Electron Microscopy Sciences, Hatfield, PA, USA). Blocks sectioned at 70 nm by ultramicrotome were post-stained with 2% aqueous uranyl acetate and 1% phosphotungstic acid, pH 3.2 [8,31]. Images were taken at 80 kV using an FEI C120 transmission electron microscope (FEI Co, Hillsboro, OR, USA) with a Gatan Orius CC Digital camera (Gatan Inc., Pleasanton, CA, USA). All images used for fibril diameter analysis, fibril density, collagen organization, and structure were taken at 45,000×. Fibril diameter distribution was visualized using ImageJ 1.54f software (National Institutes of Health, Bethesda, MD, USA) and means were calculated from 5 images per testing group within each biological sample with no more than 100 fibrils per image counted for a total of 500 fibril diameters per biological sample. Fibril density and fibrils per area of extracellular matrix (ECM) per image were calculated using the same 5 images as for the fibril diameters. Analyses of fibril diameters were carried out with one-way ANOVA with Dunnett’s multiple comparison test.

### 4.7. Dexamethasone Challenge

TP and PERI progenitor cells were thawed and plated at 1 × 10^6^ cells per T75 tissue-culture-treated flask in α-MEM, 10% fetal bovine serum, 2 mM L-Glutamine, 100 U/mL penicillin, 100 μg/mL streptomycin, and 250 ng/mL amphotericin B. Once cells reached 80% confluence, they were dissociated from flasks with trypsin and seeded at cells/cm^3^; cells were then treated with 0 nM, 5 nM, and 10 nM dexamethasone (Sigma-Aldrich, St. Louis, USA) for 3 days. After 3 days, cells were trypsinized, stained with Trypan blue, and counted. Analyses of dexamethasone challenge were performed using 2-way ANOVA. Once 10 nM dexamethasone was determined to be the optimal dose for affecting cell numbers, TP and PERI progenitor cells were again seeded at 75,000 cells/well (6-well plates) in α-MEM, 10% fetal bovine serum, 2 mM L-Glutamine, 100 U/mL penicillin, 100 μg/mL streptomycin, and 250 ng/mL amphotericin B. Cells were treated with 10 nM dexamethasone along with vitamin C concentrations of 0 μM, 200 μM, 400 μM, and 800 μM for 3 days. After three days, apoptosis was measured using the Apo-ONE^®^ Homogeneous Caspase-3/7 Assay (Promega, Madison, USA) with measurements taken using a spectrofluorometer configured at an excitation wavelength range of 485 ± 20 nm and an emission wavelength range of 530 ± 25 nm. Analyses of apoptosis assays were performed using one-way ANOVA with Dunnett’s multiple comparison test.

## 5. Conclusions

In this study, we hypothesized that vitamin C would improve tenogenic properties for equine superficial digital flexor tendon TP and PERI cells. We demonstrated that equine SDFT TP and PERI cells maintained in three-dimensional constructs benefited in many ways from the supplementation of vitamin C. Expression profiling demonstrated that vitamin C had anti-inflammatory properties and could mitigate matrix degradation, chondrification, and ossification properties for TP and PERI cells. Moreover, when supplemented with vitamin C, tendon marker genes had increasing trends for TP- and PERI-cell-derived constructs. Furthermore, collagen content increased in tendon constructs supplemented with vitamin C. Additionally, vitamin C was cytoprotective for TP cells when challenged with dexamethasone, though this was not the case with PERI cells. There were also distinct differences in the expression profiles of TP and PERI cells treated with vitamin C; in particular, genes involved in immunity and inflammatory processes generally had much lower expression in TP cells. Thus, there seems to be great potential for improving the tenogenic properties of equine TP and PERI cells via the supplementation of vitamin C. Based on these findings, future studies will further explore the effects of vitamin C on equine tendon cells, particularly in an inflammatory setting as well as in vivo to discern the impact of the tendon niche on vitamin C delivered within the tissue.

## Figures and Tables

**Figure 1 ijms-24-17098-f001:**
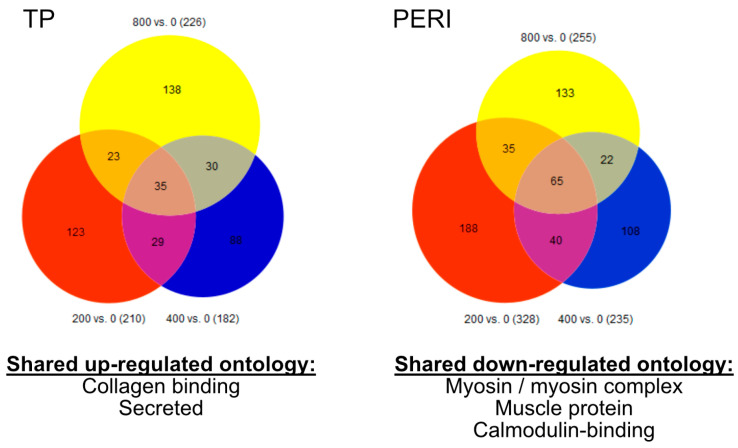
Venn diagrams demonstrate that vitamin C supplementation improved collagen organization for TP cells and reduced muscle-like properties for PERI cells. Venn diagrams show distribution of genes in comparisons of TP- and PERI-derived constructs with various supplementation amounts (vitamin C) (*p* < 0.05). The central number represents genes shared across all comparisons. When applied to the Functional Annotation program in The Database for Annotation, Visualization and Integrated Discovery (DAVID), shared functions arose.

**Figure 2 ijms-24-17098-f002:**
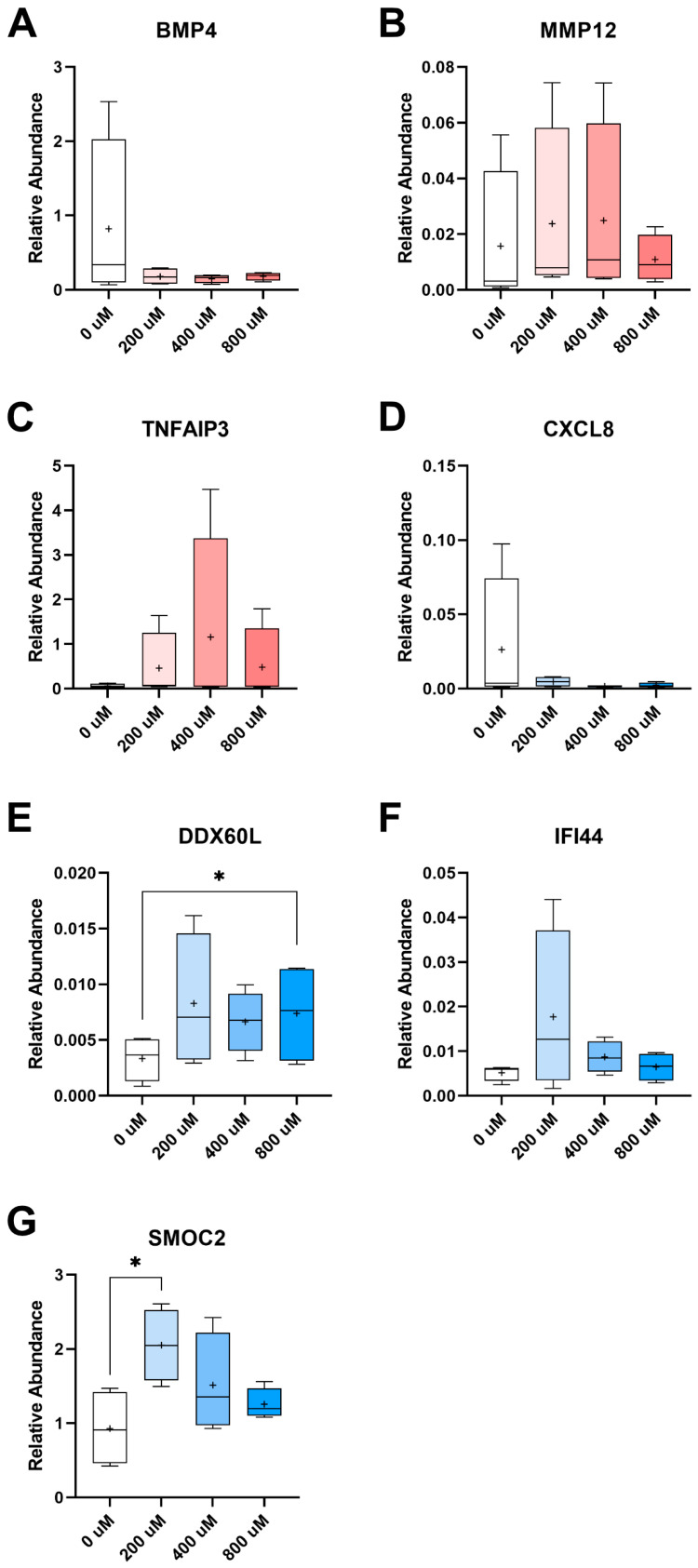
RT-qPCR confirmed significant and trending differential expression. PERI genes *BMP4* (**A**), *MMP12* (**B**), and *TNFAIP3* (**C**) demonstrated similar trends to those seen in RNAseq profiling. Likewise, profiling trends matched for TP genes *CXCL8* (**D**), *DDX60L* (**E**), *IFI44* (**F**), and *SMOC2* (**G**). Vitamin C concentrations examined are listed along the *x*-axis. The “+” within each box plot represents the mean; the line within each box represents the median. Statistical significance according to one-tailed Freidman’s non-parametric test with Dunn’s multiple test correction (*, *p* < 0.05, *n* = 4).

**Figure 3 ijms-24-17098-f003:**
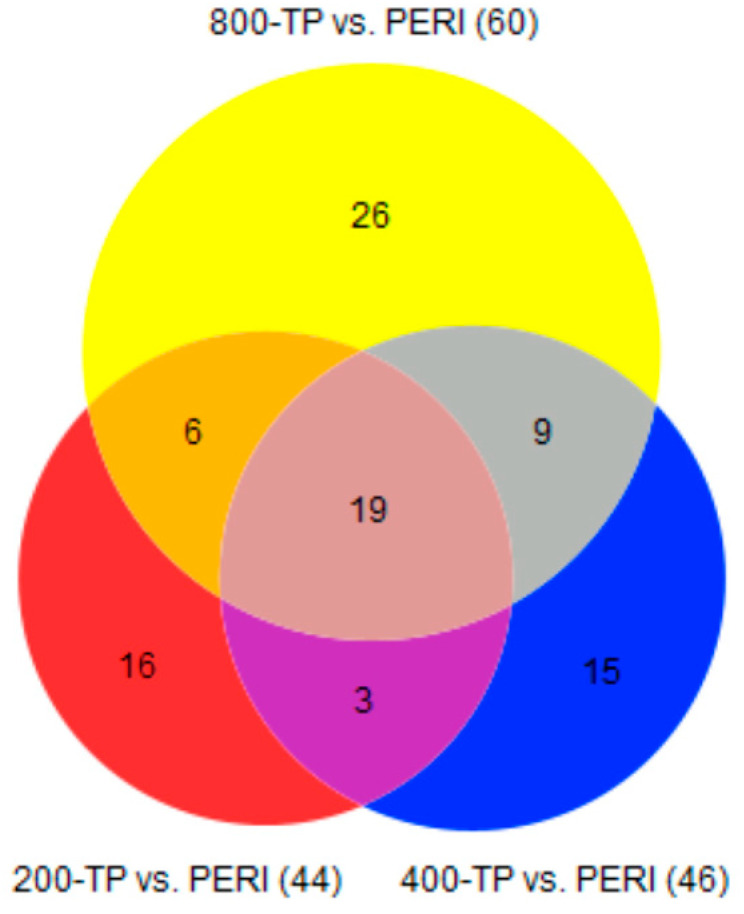
Venn diagrams demonstrate that vitamin C supplementation affected TP vs. PERI comparisons. Venn diagrams show distribution of genes in comparisons of TP- vs. PERI-derived constructs at three concentrations of vitamin C (*q* ≤ 0.05). The central number represents the number of genes shared across all TP vs. PERI comparisons.

**Figure 4 ijms-24-17098-f004:**
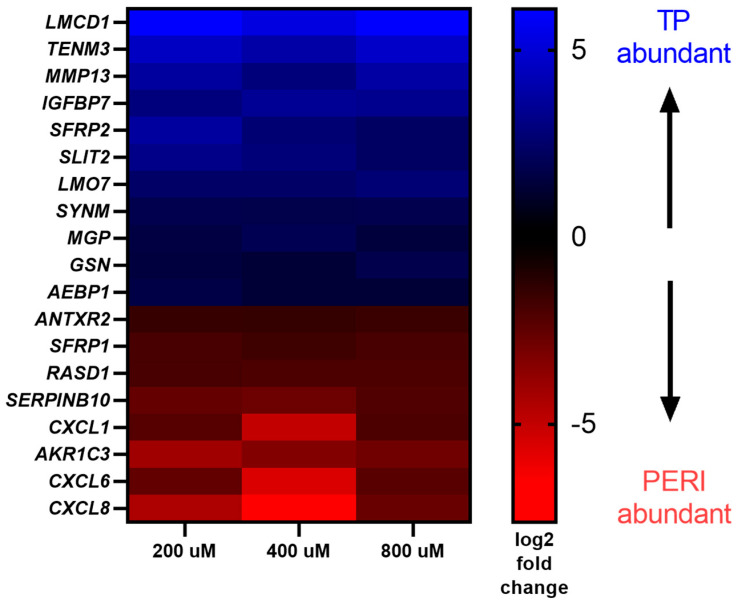
A heatmap of the genes that demonstrate differential expression between TP and PERI cells when supplemented with vitamin C. At all three concentrations of vitamin C, 19 genes consistently demonstrated a fold change difference between TP- and PERI-derived constructs (*q* ≤ 0.05).

**Figure 5 ijms-24-17098-f005:**
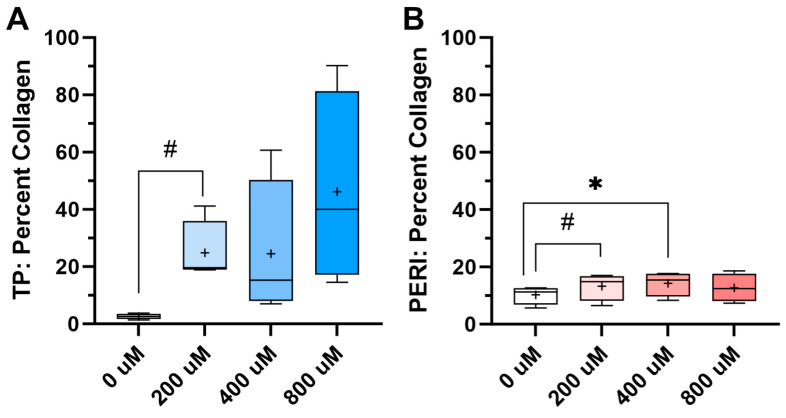
Assays for collagen content (provided as % dry weight) indicated that vitamin C supplementation increased the amount of collagen deposited in the constructs. Assays for collagen content indicated that vitamin C supplementation improved the amount of collagen deposited in the constructs. Tendon proper (TP) cells (**A**) deposited collagen with 200 μM vitamin C supplementation, while peritenon (PERI) cells (**B**) deposited more collagen with 200 μM and 400 μM vitamin C supplementation. One-way ANOVA with Dunnett’s multiple comparison test. Significance is reported as * = *p* < 0.05 and # = 0.05 < *p* < 0.10. Mean shown as “+” (*n* = 4).

**Figure 6 ijms-24-17098-f006:**
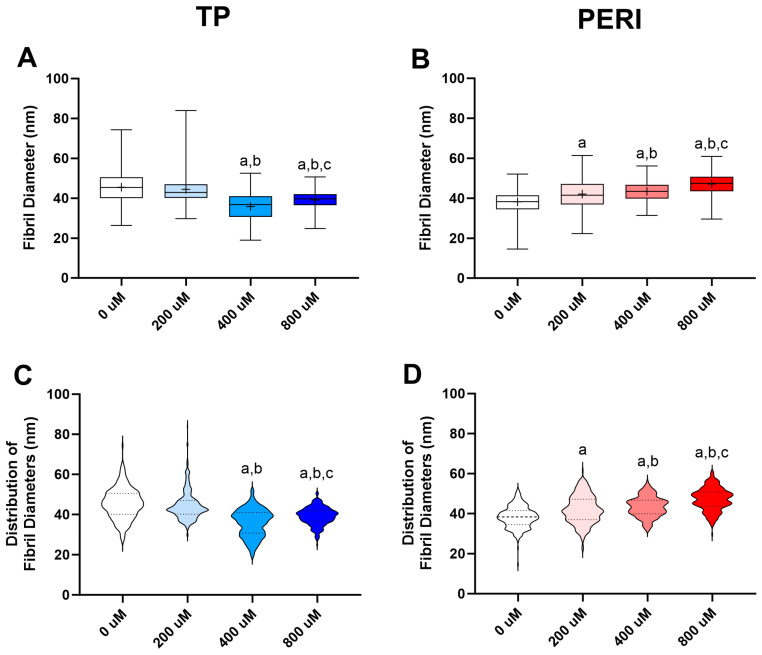
TEM fibril diameter distributions. Fibril diameter analysis for samples supplemented with 0 μM, 200 μM, 400 μM, and 800 μM vitamin C. Fibril diameter distributions are given as box plots and violin plots for constructs seeded with tendon proper (TP) cells (**A**,**C**) and peritenon (PERI) cells (**B**,**D**). Analyses of fibril diameters were carried out with one-way ANOVA with Dunnett’s multiple comparison test (*n* = 4). Significance: a, vs. 0 μM; b, vs. 200 μM, c, vs. 400 μM. Examples of TEM images used in measuring fibril diameters can be found in Appendix A.

**Figure 7 ijms-24-17098-f007:**
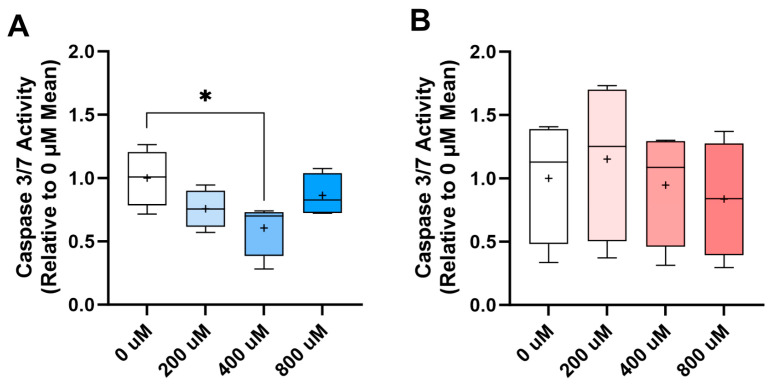
Apoptotic activity reduced for dexamethasone-challenged TP cells supplemented with vitamin C. TP (**A**) and PERI (**B**) cells were challenged for three days with 10 nM dexamethasone; increasing concentrations of vitamin C were applied. Apoptosis reduced for TP cells supplemented with vitamin C. Results are displayed as Caspase 3/7 activity of cells at each concentration of vitamin C relative to 0 μM vitamin C control. Mean shown as “+”. Paired one-way ANOVA with Dunnett’s multiple comparison test (significance: *, *p* < 0.05, *n* = 4).

**Table 1 ijms-24-17098-t001:** Reads mapped to EquCab3 horse genome.

Horse ID	TP Constructs Mean Reads	TP Constructs Mean % Mapped	PERI Constructs Mean Reads	PERI Constructs Mean% Mapped
5	5.21 million	94.4%	5.07 million	94.0%
6	5.25 million	94.2%	4.91 million	94.7%
7	5.06 million	94.1%	5.07 million	91.7%
10	4.86 million	94.5%	5.10 million	95.1%

Note: Horse ID is the laboratory’s identifier for specific horses from which cells were isolated.

**Table 2 ijms-24-17098-t002:** Differentially expressed genes: comparing vitamin C concentrations (*p* < 0.05 and *q* < 0.20).

Compared to 0 μM	TP *p* < 0.05	TP *q* < 0.20	PERI *p* < 0.05	PERI *q* < 0.20
200 μM vs. 0 μM	210	3	328	8
400 μM vs. 0 μM	182	4	235	0
800 μM vs. 0 μM	226	1	255	2

**Table 3 ijms-24-17098-t003:** Differential expression of genes from 3′-Tag-seq analyses.

Gene	TP/ PERI	[Vit C]	Fold Change	*p*-Value	*q*-Value	Annotation
*ADAMTS1*	PERI	200 vs. 0	−4.20	1.28 × 10^−4^	0.12	Encodes a proteoglycanase
*BMP4*	PERI	200 vs. 0	−6.50	1.01 × 10^−5^	0.03	Promotes calcification and ossification
*CCL13*	PERI	200 vs. 0	−7.71	1.34 × 10^−5^	0.03	Pro-inflammatory protein
*GPC3*	PERI	200 vs. 0	4.72	2.02 × 10^−5^	0.03	Expressed at tendon insertion
*IL11*	PERI	200 vs. 0	−3.27	5.89 × 10^−5^	0.08	Fibrotic tissue marker
*IRF8*	PERI	200 vs. 0	−21.95	1.96 × 10^−4^	0.17	Pro-inflammatory protein
*MMP12*	PERI	800 vs. 0	−6.50	7.92 × 10^−6^	0.05	Encodes degradative protein
*MYH7*	PERI	200 vs. 0	−45.62	8.31 × 10^−6^	0.03	Encodes cardiac and skeletal myosin protein
	PERI	800 vs. 0	−50.46	4.72 × 10^−6^	0.05	
*TNFAIP3*	PERI	200 vs. 0	32.01	8.71 × 10^−5^	0.10	Anti-inflammatory protein
*CXCL6*	TP	400 vs. 0	−6.80	1.42 × 10^−4^	0.14	Pro-inflammatory marker
*CXCL8*	TP	200 vs. 0	−15.50	2.58 × 10^−5^	0.11	GAG-binding pro-inflammatory marker
	TP	400 vs. 0	−14.37	1.19 × 10^−4^	0.14	
*DDX60L*	TP	200 vs. 0	7.46	6.41 × 10^−6^	0.04	DExD/H-box helicase protein
*IFI44*	TP	200 vs. 0	6.42	3.77 × 10^−7^	4.78 × 10^−3^	Inhibits calcification
*JAM2*	TP	800 vs. 0	6.86	1.68 × 10^−5^	0.10	Proliferation/migration marker
*SAA1*	TP	400 vs. 0	−4.93	2.17 × 10^−5^	0.09	Active inflammation marker
*SMOC2*	TP	400 vs. 0	2.34	4.97 × 10^−5^	0.10	Promotes matrix assembly

Note: positive fold differences represent greater expression in vitamin C-supplemented constructs, while negative fold differences represent greater expression in control constructs.

**Table 4 ijms-24-17098-t004:** Differentially expressed genes: comparing concentrations (*p* < 0.05 and *q* < 0.20).

Comparing TP vs. PERI at Different Vitamin C Concentrations	TP vs. PERI *q* ≤ 0.05	DAVID Functional Annotation Enriched Categories (FDR < 0.05)
0 μM	18	TP: n/a
	8	PERI: extracellular space (CC)
200 μM	31	TP: extracellular space (CC), calcium ion binding (MF), extracellular region (CC)
	13	PERI: CXCR chemokine receptor binding (MF), chemokine-mediated signaling pathway (BP), chemokine activity (MF), antimicrobial humoral immune response mediated by antimicrobial peptides (BPs), neutrophil chemotaxis (BP), heparin binding (MF)
400 μM	21	TP: n/a
	25	PERI: extracellular space (CC), chemokine activity (MF), CXCR chemokine receptor binding (MF), chemokine-mediated signaling pathway (BP), heparin binding (MF), growth factor activity (MF), cellular response to interleukin-1 (BP), neutrophil chemotaxis (BP), positive regulation of peptidyl-serine phosphorylation (BP), cellular response to tumor necrosis factor (BP), inflammatory response (BP), cytokine activity (MF), positive regulation of vascular smooth muscle cell proliferation (BP), cellular response to lipopolysaccharide (BP), positive regulation of cell proliferation (BP), extracellular region (CC)
800 μM	39	TP: extracellular region (CC)
	21	PERI: extracellular space (CC), extracellular region (CC), CXCR chemokine receptor binding (MF), heparin binding (MF), growth factor activity (MF), chemoattractant activity (MF)

DAVID: Database for Annotation, Visualization and Integrated Discovery; FDR: false discovery rate; CC: cellular compartment; MF: molecular function; BP: biological process.

## Data Availability

The datasets used and/or analyzed during the current study are available from the corresponding author on reasonable request. RNAseq data files (fastq) and their metadata are available at the Sequence Read Archive (SRA) of the NIH National Library of Medicine at the accession number PRJNA998601.

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
