# Peer review of "Examining the Potential of Vitamin C Supplementation in Tissue-Engineered Equine Superficial Digital Flexor Tendon Constructs"

_ijms, 2023, doi:10.3390/ijms242317098_

Round 1

Reviewer 1 Report

Comments and Suggestions for Authors

In this study the authors analyzed the effect of vitamin C on tendon and peritenon cells in order to evaluate the eventual improvement in tenogenic properties. Gene expression profiles of the constructs were assessed and collagen fibril content and ultrastructure were also analyzed. Moreover, also the antiapototic effect of vitamin C was assayed. The results suggest ant anti- inflammatory effect on cell constructs.

The topic of this study is interesting and the mauscript is clear but with several areas of weakness.

The first issue is that the overall effect of vitamin C resulted mostly related to the decrease of inflammation and inhibition of apoptosis, but no evidence of an effect on collagen and collagen turnover pathways was demonstrated. To understand the effect on tenogenic ability of tendon cells collagen secretion in the cell culture medium and the MMP activity should be analzed as well, including also the quantification of TIMPs.

Moreover, the increased collagen content in samples treated with vitamin C compared to samples cultured in absence of vitamin C in not relevant and surprising since it is well know that collagen synthesis needs vitamin C as cofactor and that for in vitro studies vitamin C is an important supplement of the cell culture medium for the analysis of collagen content and turnover.

Finally, it is not clear if in this study were used tendon and peritenon cells or their progenitors. If progenitor cells were used, the rationale for their use instead of tenocytes able for tendon repair should be clearly explained. Furthermore, if tendon progenitor cells were used, their phenotype different from that of tenocytes should be clearly demonstrated.

Specific comments:

Introduction, line 46: please change “tendon” into “tendon connective tissue”.

Introduction, lines 53-54: ascorbic acid is an essential cofactor required also for the hydroxylation of lysine to hydroxylysine.

Discussion, lines 254-256: the authors should consider that vitamin C, in their experimental setting, does not promote the secretion of extracellular molecules related to collagen. Indeed, the analyzed molecules are not related to collagen and they were analyzed only at the level of their gene expression and not at the protein level in cell culture medium.

Discussion, lines 312-314: the authors suggest that “…., the fibril diameters of PERI cell-derived increased in diameter which suggests that the fibrils generated were maturing to thicker fibril diameter sizes over the period of culture”. I think that this hypothesis is inconsistent if not demonstrated by analyzing LH and LOX gene expression in relation to the role of LOX and LH to promote collagen maturation by crosslinking.

Materials and Methods, line 360: tendon samples should be corrected into tendon cells. Moreover, the construct used for the growth of cells should be described in detail.

Supplement: COL1A1 is not a tendon marker but is a general marker of fibroblasts and mesenchymal cells.

Author Response

Responses to Reviewer #1's Comments:

In this study the authors analyzed the effect of vitamin C on tendon and peritenon cells in order to evaluate the eventual improvement in tenogenic properties. Gene expression profiles of the constructs were assessed and collagen fibril content and ultrastructure were also analyzed. Moreover, also the antiapototic effect of vitamin C was assayed. The results suggest an anti- inflammatory effect on cell constructs.

The topic of this study is interesting and the manuscript is clear but with several areas of weakness.

The authors: We thank the reviewer for their critiques and hope that they decide that the revised version of the manuscript addresses the weaknesses that they’ve described.

The first issue is that the overall effect of vitamin C resulted mostly related to the decrease of inflammation and inhibition of apoptosis, but no evidence of an effect on collagen and collagen turnover pathways was demonstrated. To understand the effect on tenogenic ability of tendon cells collagen secretion in the cell culture medium and the MMP activity should be analzed as well, including also the quantification of TIMPs.

Moreover, the increased collagen content in samples treated with vitamin C compared to samples cultured in absence of vitamin C in not relevant and surprising since it is well know that collagen synthesis needs vitamin C as cofactor and that for in vitro studies vitamin C is an important supplement of the cell culture medium for the analysis of collagen content and turnover.

The authors: We certainly agree with the reviewer about the importance of vitamin C to collagen synthesis. Thus, by measuring collagen content in the presence of vitamin C, we are confirming the responsiveness of both TP and PERI cell types to the supplemented vitamin C. The cells are in three-dimensional in vitro constructs that are originally composed of fibrinogen and thrombin. Any collagen found within the constructs represents collagen secreted by the cells into the fibrin fibrils. This is a well-established in vitro model. We have added text to the revised manuscript to clarify that collagen deposited in the constructs comes from the cells (Lines 197-200). Regrettably, we did not measure MMP and TIMP activity using zymography or biochemical assays. We have added a statement in a paragraph on study limitations indicating that the measurement of MMP and TIMP activity would be of great value, particularly in follow-up we plan to do in which constructs will also be challenged with inflammatory conditions (Lines 372-376).

Finally, it is not clear if in this study were used tendon and peritenon cells or their progenitors. If progenitor cells were used, the rationale for their use instead of tenocytes able for tendon repair should be clearly explained. Furthermore, if tendon progenitor cells were used, their phenotype different from that of tenocytes should be clearly demonstrated.

The authors: We used a strategy originally developed by Bi et al. and used previously in our laboratory for mouse tendon progenitors. More context is now provided in the Introduction (Lines 45-54), Discussion (Lines 376-386), Materials & Methods (Lines 410-420), and Figure S4, where cells were phenotyped for markers as described by Lui (2015), Bundgaard et al. (2018), and Li et al. (2021).

Specific comments:

Introduction, line 46: please change “tendon” into “tendon connective tissue”.

The authors: This revision has been made (Line 49).

Introduction, lines 53-54: ascorbic acid is an essential cofactor required also for the hydroxylation of lysine to hydroxylysine.

The authors: This revision has been made (Lines 59-61).

Discussion, lines 254-256: the authors should consider that vitamin C, in their experimental setting, does not promote the secretion of extracellular molecules related to collagen. Indeed, the analyzed molecules are not related to collagen and they were analyzed only at the level of their gene expression and not at the protein level in cell culture medium.

The authors: The reviewer raises an excellent point! We have corrected this statement to only reflect the gene expression changes (Lines 268-281).

Discussion, lines 312-314: the authors suggest that “…., the fibril diameters of PERI cell-derived increased in diameter which suggests that the fibrils generated were maturing to thicker fibril diameter sizes over the period of culture”. I think that this hypothesis is inconsistent if not demonstrated by analyzing LH and LOX gene expression in relation to the role of LOX and LH to promote collagen maturation by crosslinking.

The authors: Transmission electron microscopy visualization of the cross-sections of the tendon constructs allows one to measure the collagen fibril diameters. In order for the collagen fibrils to exist and then have visible diameters, they must assemble, which requires cross-linking. Thus, because there are fibrils in a construct that previously had no collagen, since all collagen is produced and deposited by the cells, LH and LOX activity must be ongoing. No differential expression was seen in the RNAseq data for LOX.

Materials and Methods, line 360: tendon samples should be corrected into tendon cells. Moreover, the construct used for the growth of cells should be described in detail.

The authors: Actual tendon samples were harvested from horses. From those tendon samples, cells were isolated. More information about the construct composition has been added (Lines 389-420).

Supplement: COL1A1 is not a tendon marker but is a general marker of fibroblasts and mesenchymal cells.

The authors: We agree that COL1A1 alone is not a tendon marker. However, COL1A1 expression is often used with several other extracellular matrix genes to assess tenogenic capabilities of cells. The legend for Figure S1 has been revised.

Reviewer 2 Report

Comments and Suggestions for Authors

I really liked this manuscript, it provides useful information for the recovery of tendon injuries, with high potential for transfer to the equine clinic. No flaws were detected and I have no suggestions to improve the article. I think it is clear, well written and the conclusions are based on the results.

Author Response

The authors: We are grateful for the critical review of the manuscript and thank the reviewer for their favorable comments about the study.

Reviewer 3 Report

Comments and Suggestions for Authors

Vitamin C is still since many decades under investigation in regard to its effects on fibroblasts. What the study makes unique is that detailed expression data are provided for major cell populations and that it is performed with equine cells. The manuscript is clearly organized and written in a well understandable manner. Some information as stated below should be added.

line 71: "improvement of tenogenic properties" should be specified.

The rationale for selecting the ascorbic acid concentrations should be mentioned (line 83), relation to normal intake and serum levels in horses and humans?

Table 1, first column, the horse numbers should be explained

Heading of table 2: add vitamin C

Table 4: shift "TP: n/a" to the left side as before

Figure 5: I would prefer to show the same Y axis maximum in A) and B). provide the number of independent experiments as in the other figure legends (fig. 2)

fig. 2+5 could also be shown as box plots

percent of what? dry weigth or construct wet weight? legend: do not use "improved" since it is not scientific

Fig. 6 TEM, exemplary representative images would be helpful. the legend talks about diameter, the Y axis is labelled with "length" what is correct? add in the legend and method section at which timepoint analyses were done.

the explanation at the X axis "vitamin concentration" could be omitted as at the other figures.

Fig. 7: "net fluorescence" what does it really mean? how many cells of the population die by apoptosis? compare line 244: decreases in surviving cells

line 266: TNF-a, write alpha

line 257: "encoding muscle proteins" how about alpha smooth muscle actin? compare line 275, insert "muscle" after smooth

line 315: how bout the enzymes responsible for crosslinking collagen? lysyl oxidase

line 352: gender and race of the horses

line 372: explain the different and low cell numbers

line 511: how should it be administered? locally or per os?  Which concentration could be promising? in cell culture, too high concentrations of vitamin C can be harmful for cells.

Author Response

Response to Reviewer #3's Comments:

Vitamin C is still since many decades under investigation in regard to its effects on fibroblasts. What the study makes unique is that detailed expression data are provided for major cell populations and that it is performed with equine cells. The manuscript is clearly organized and written in a well understandable manner. Some information as stated below should be added.

The authors: We are grateful for the reviewer’s comments and the critiques and suggestions that follow. We will address each concern individually.

line 71: "improvement of tenogenic properties" should be specified.

The authors: We added a sentence after the hypothesis (Lines 77-78).

The rationale for selecting the ascorbic acid concentrations should be mentioned (line 83), relation to normal intake and serum levels in horses and humans?

The authors: In the Introduction we added an explanation for the doses of vitamin C added, as well as described the plasma levels of ascorbic acid found in adult horses and age-adjusted mean concentrations for humans (Lines 88-92).

Table 1, first column, the horse numbers should be explained

The authors: This has been changed to Horse ID with a clarifying footnote (Lines 113-115).

Heading of table 2: add vitamin C

The authors: “Vitamin C” has been added as suggested.

Table 4: shift "TP: n/a" to the left side as before

The authors: The “n/a” has been shifted.

Figure 5: I would prefer to show the same Y axis maximum in A) and B). provide the number of independent experiments as in the other figure legends (fig. 2)

The authors: The figure and its legend have been revised as requested.

fig. 2+5 could also be shown as box plots

The authors: The figure has been revised as requested.

percent of what? dry weigth or construct wet weight? legend: do not use "improved" since it is not scientific

The authors: Percent dry weight has been added to the legend. “Improved” has been changed to “increased.”

Fig. 6 TEM, exemplary representative images would be helpful. the legend talks about diameter, the Y axis is labelled with "length" what is correct? add in the legend and method section at which timepoint analyses were done.

The authors: We have corrected the labels for the Y axes. Some example TEM images are provided in Figure S3.

the explanation at the X axis "vitamin concentration" could be omitted as at the other figures.

The authors: We have made this revision as requested.

Fig. 7: "net fluorescence" what does it really mean? how many cells of the population die by apoptosis? compare line 244: decreases in surviving cells

The authors: We changed the figure format, removing net fluorescence, and instead normalized the fluorescence to the mean apoptotic activity of dexamethasone-treated cells receiving no vitamin C. Also, we clarified that we counted cell numbers for Figure S2 and for Figure 7 we actually measured apoptotic activity to understand if apoptosis accounted for loss of cells. In the apoptotic assay, you do not count cells but instead measure apoptotic activity. To clarify, Figure S2 represents cell numbers but not necessarily apoptosis as it was not measured – instead just cell numbers. Furthermore, Figure 7 represents apoptotic activity, which could represent ultimate fate of cell death via apoptosis specifically.

line 266: TNF-a, write alpha

The authors: The “a” has been replaced with “alpha” (Now Line 285).

line 257: "encoding muscle proteins" how about alpha smooth muscle actin? compare line 275,

The authors: When looking at overrepresentation, the genes encoding muscle proteins can be found in Table S6. Alpha smooth muscle actin was not in the table.

insert "muscle" after smooth

The authors: “Muscle” has been added.

line 315: how bout the enzymes responsible for crosslinking collagen? lysyl oxidase

The authors: Significant differences in transcript numbers for lysyl oxidase were not noted between constructs treated with difference concentrations of vitamin C. We did not measure lysyl oxidase activity outside of RNAseq. The assembly of fibrils of detectable diameters is at least indirectly indicative of activity of LOX because the assembly of fibrils and thus increased fibril diameters would depend upon LOX activity.

line 352: gender and race of the horses

The authors: It is now listed: “Thoroughbreds and Quarterhorses ages 8-14 years, 2 mares and 2 stallions.”

line 372: explain the different and low cell numbers

The authors: Low cell seeding densities are required to encourage adhering progenitor cells in culture to form colonies of clones. Citations to the technique have also been added to point readers toward previous studies using this technique (Lines 410-412).

line 511: how should it be administered? locally or per os?  Which concentration could be promising? in cell culture, too high concentrations of vitamin C can be harmful for cells.

The authors: That is certainly a great question! We would prefer not to speculate further on this within the manuscript. It likely would not be orally as others have shown that there are peak levels that plateau in the plasma when supplemented. Those levels do not reach the 200μM-800μM concentrations. Likely something provided more locally to the tendon would have the best effect, but again this is speculation at this point.

Round 2

Reviewer 1 Report

Comments and Suggestions for Authors

The revised manuscript is greatly improved and most issues were addressed. It can be accepted for publication.

Author Response

The authors are very grateful for the reviewer's valuable critiques as they strengthened the paper.

Reviewer 3 Report

Comments and Suggestions for Authors

My comments have been sufficiently addressed by the revision of the authors.

Author Response

(The authors gave the same response as above.)
